# Volatile Organic Compounds Emitted by the Biocontrol Agent *Pythium oligandrum* Contribute to Ginger Plant Growth and Disease Resistance

Taha Majid Mahmood Sheikh,[a] Dongmei Zhou,[a] Haider Ali,[b] Sarfraz Hussain,[c] Nan Wang,[a] Siqiao Chen,[a,d] Yishen Zhao,[a,e] Xian Wen,[a,e] Xiaoyu Wang,[a] Jinfeng Zhang,[a] Lunji Wang,[e] Sheng Deng,[a] Hui Feng,[a] Waseem Raza,[f] Pengxiao Fu,[g] Hao Peng,[g] Lihui Wei,[a] Paul Daly[a]

[a]Key Lab of Food Quality and Safety of Jiangsu Province—State Key Laboratory Breeding Base, Institute of Plant Protection, Jiangsu Academy of Agricultural Sciences, Nanjing, China
[b]School of Biosciences, University of Birmingham, Birmingham, United Kingdom
[c]Key Laboratory of Integrated Regulation and Resource Development on Shallow Lakes of Ministry of Education, College of Environment, Hohai University, Nanjing, China
[d]Fungal Genomics Laboratory (FungiG), Jiangsu Provincial Key Lab of Organic Solid Waste Utilization, Nanjing Agricultural University, Nanjing, China
[e]College of Food and Bioengineering, Henan University of Science and Technology, Luoyang, Henan, China
[f]Jiangsu Provincial Key Lab for Organic Solid Waste Utilization, National Engineering Research Center for Organic-based Fertilizers, Jiangsu Collaborative Innovation Center for Solid Organic Waste Resource Utilization, Nanjing Agricultural University, Nanjing, China
[g]Jiangsu Coastal Ecological Science and Technology Development Co., Ltd., Nanjing, China

**ABSTRACT** The oomycete *Pythium oligandrum* is a potential biocontrol agent to control a wide range of fungal and oomycete-caused diseases, such as *Pythium myriotylum*-caused rhizome rot in ginger, leading to reduced yields and compromised quality. Previously, *P. oligandrum* has been studied for its plant growth-promoting potential by auxin production and induction of disease resistance by elicitors such as oligandrin. Volatile organic compounds (VOCs) play beneficial roles in sustainable agriculture by enhancing plant growth and resistance. We investigated the contribution of *P. oligandrum*-produced VOCs on plant growth and disease suppression by initially using *Nicotiana benthamiana* plants for screening. *P. oligandrum* VOCs significantly enhanced tobacco seedling and plant biomass contents. Screening of the individual VOCs showed that 3-octanone and hexadecane promoted the growth of tobacco seedlings. The total VOCs from *P. oligandrum* also enhanced the shoot and root growth of ginger plants. Transcriptomic analysis showed a higher expression of genes related to plant growth hormones and stress responses in the leaves of ginger plants exposed to *P. oligandrum* VOCs. The concentrations of plant growth hormones such as auxin, zeatin, and gibberellic acid were higher in the leaves of ginger plants exposed to *P. oligandrum* VOCs. In a ginger disease biocontrol assay, the VOC-exposed ginger plants infected with *P. myriotylum* had lower levels of disease severity. We conclude that this study contributes to understanding the growth-promoting mechanisms of *P. oligandrum* on ginger and tobacco, priming of ginger plants against various stresses, and the mechanisms of action of *P. oligandrum* as a biocontrol agent.

**IMPORTANCE** Plant growth promotion plays a vital role in enhancing production of agricultural crops, and *Pythium oligandrum* is known for its plant growth-promoting potential through production of auxins and induction of resistance by elicitors. This study highlights the significance of *P. oligandrum*-produced VOCs in plant growth promotion and disease resistance. Transcriptomic analyses of leaves of ginger plants exposed to *P. oligandrum* VOCs revealed the upregulation of genes involved in plant growth hormone signaling and stress responses. Moreover, the concentration of growth hormones significantly increased in *P. oligandrum* VOC-exposed ginger plants. Additionally, the disease severity was reduced in *P. myriotylum*-infected ginger plants exposed to *P. oligandrum* VOCs. In ginger, *P. myriotylum*-caused rhizome rot disease results in severe losses, and biocontrol has a role as

Address correspondence to Lihui Wei, weilihui@jaas.ac.cn, or Paul Daly, paul.daly@jaas.ac.cn.

The authors declare a conflict of interest. L.W. and D.Z. are co-authors on a patent application for *P. oligandrum* (Patent application number CN201910757035.2) relating to the biocontrol of plant diseases using the *P. oligandrum* GAQ1 isolate.

[This article was published on 3 August 2023 with an error in the supplemental material. Table S2 was corrected in the current version, posted on 17 August 2023.]

part of an integrated pest management strategy for rhizome rot disease. Overall, growth enhancement and disease reduction in plants exposed to *P. oligandrum*-produced VOCs contribute to its role as a biocontrol agent.

**KEYWORDS** *Pythium oligandrum*, volatile organic compounds, ginger, *Zingiber officinale*, transcriptomic analysis, plant growth promotion, *Pythium myriotylum*, disease severity

Plant growth promotion by beneficial microorganisms is a well-studied phenomenon frequently linked to the production of microbial volatile organic compounds (VOCs) and phytohormones (1–4). *Pythium oligandrum*, a non-plant-pathogenic oomycete, has been studied for its antagonistic effects against plant pathogens (5, 6) as well as reported for its plant growth-promoting potential, either by the production of compounds such as auxins (7) or by induction of resistance through elicitors (8). Previously, *P. oligandrum* VOCs have been suggested to contribute to the growth inhibition of the plant pathogens *Phoma medicaginis*, *Mycosphaerella pinodes* (9), *Pythium ultimum*, and *Fusarium oxysporum* (10). Moreover, we found that *P. oligandrum*-produced VOCs inhibit *Pythium myriotylum* (11). *P. myriotylum* is a broad-host-range plant-pathogenic oomycete that causes a range of soilborne diseases, such as ginger soft rot disease (12–14), damping-off of chili pepper (15), and crown and root rot of hemp (16).

VOCs produced by antagonistic bacteria (1) and fungi (17) have been studied for their potential for plant growth promotion and regulation of hormone synthesis and nutrient transport (18–20). VOCs produced by *Bacillus subtilis* promoted *Arabidopsis thaliana* growth via the cytokinin signaling pathway (21) and by regulating the expression of 3-indoleacetic acid (IAA)-related genes (22). The VOC 2*R*,3*R*-butanediol produced by *Pseudomonas chlororaphis* O6 induced drought tolerance in *A. thaliana* through a salicylic acid (SA) signaling pathway-dependent mechanism by regulating stomatal opening and closing (23). *Trichoderma asperellum* VOCs promoted the growth of lettuce and induced resistance against *Corynespora cassiicola* and *Curvularia aeria* (17). In *Nicotiana benthamiana* seedlings exposed to the VOCs produced by *Microbacterium aurantiacum*, there was upregulated expression of the key genes involved in auxin, ethylene, jasmonic acid, and salicylic acid pathways (24). A volatile compound, tridecane, produced by *Bacillus polymyxa* E681 induced systemic resistance in *Arabidopsis* plants (25). *Streptomyces* spp. produce 3-octanone, which has been reported to enhance root growth in *Arabidopsis* (26). Hexadecane promoted plant growth and induced systemic resistance in *Arabidopsis* plants against *Pectobacterium carotovorum* and *Pseudomonas syringae* (27).

Oomycete diseases are among the most devastating plant diseases, causing losses of billions of dollars annually in agriculture (28, 29). *P. myriotylum* has been reported to cause diseases in a wide range of crops (30), including tobacco (31) and ginger (13). Pythium soft rot caused by *P. myriotylum* results in significant losses in crop yields (32, 33). As well as *P. oligandrum*, other species of non-plant-pathogenic oomycetes have been reported as antagonists of fungal and oomycete plant pathogens and potential biocontrol agents. For example, *Pythium nunn* has been reported as a parasite of an oomycete plant pathogen, *Pythium ultimum* (34), and *Pythium periplocum* antagonized the fungal plant pathogen *Botrytis cinerea* (35). Studies have revealed that oomycete species belonging to the *Phytophthora* or *Pythium* genus emit VOCs containing a diverse array of chemical groups, such as esters, alcohols, ketones, and alkenes. Interestingly, different plant-pathogenic *Phytophthora* species have been found to produce distinct sets of VOCs. For example, *P. plurivora* produces acetoin, 4-hydroxybutanoic acid, $\alpha$-pinene, and $\Delta$-3-carene, while *P. cactorum* produces acetone, dimethyl-disulfide, 1-hexanol, 1-heptanol, 1-octen-3-ol, 3-octanone, and 2-octen-1-ol (36). For *Pythium* species, the microbial antagonist *P. oligandrum* has been reported in our previous study to produce a range of VOCs, including methyl heptenone, 2-undecanone, 3-octanone, octanal, and hexadecane (11).

In our previous study, we demonstrated that *P. oligandrum*-produced VOCs could inhibit the growth of the ginger pathogen *P. myriotylum*. In this study, we aimed to investigate if

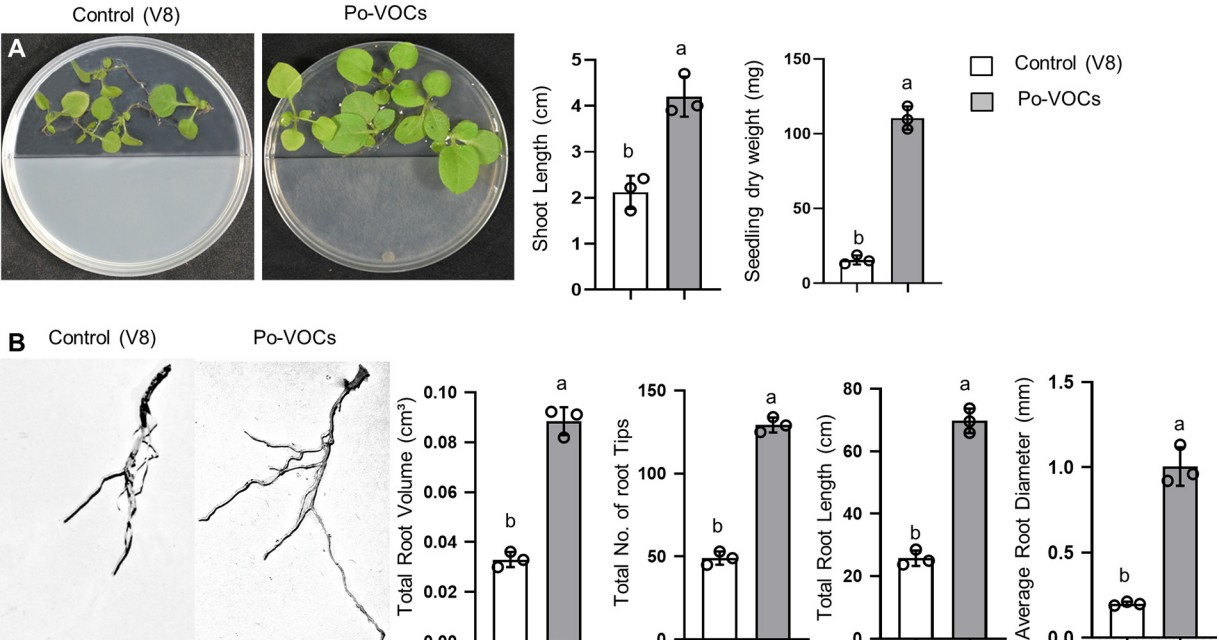

**FIG 1** *P. oligandrum* volatile organic compounds (Po-VOCs) enhanced the growth of *Nicotiana benthamiana* seedlings. (A) The plates were imaged 10 days after exposure to Po-VOCs or the control (V8), and shoot length and seedling dry weight were measured. (B) Representative images of the roots of *N. benthamiana* seedlings exposed to Po-VOCs or control (V8) for 10 days. The scale bar represents 2 cm. Four root morphological parameters, including total root volume, total number of root tips, total root length, and average root diameter, were measured. Error bars indicate the standard error of the mean ($n = 3$). Different lowercase letters above the bars represent significant differences between treatments. Experiments were repeated twice with similar results, as shown in Fig. S1.

*P. oligandrum* VOCs could contribute to plant growth promotion and disease control apart from the previously reported auxin-mediated mechanism of growth promotion and elicitor-induced disease resistance in plants. Initially, *N. benthamiana* was used to screen the effects as well as for deconvoluting the total VOCs. We found support for the role of *P. oligandrum*-produced VOCs in plant growth promotion and disease resistance. Transcriptome and metabolite analyses suggested that ginger growth promotion was mediated by changes in hormone levels.

## RESULTS

***P. oligandrum* volatile organic compounds enhanced seedling growth in *N. benthamiana in vitro*.** To initially investigate the growth promotion potential of *P. oligandrum* GAQ1 VOCs (Po-VOCs) on *N. benthamiana* seedlings, a bipartite petri plate was used to prevent diffusion of secreted growth-promoting compounds through the agar. After 10 days, there were significant increases in shoot length (2.25-fold) and seedling dry weight (5.5-fold) in the seedlings exposed to Po-VOCs compared to the control (Fig. 1A; see Fig. S1 in the supplemental material). Moreover, roots were scanned to analyze the morphology of the roots exposed to Po-VOCs (Fig. 1B). The root morphological parameters showed that there were significant increases in the total root volume (3-fold), total number of root tips (2.6-fold), total root length (3-fold), and average root diameter (5-fold) of the seedlings exposed to Po-VOCs compared to the control (Fig. 1B).

**Exposure to *P. oligandrum* VOCs increased the growth of *N. benthamiana in planta*.** *In planta* assays were performed to determine the growth-promoting potential of Po-VOCs in *N. benthamiana* plants. *N. benthamiana* plants were grown in the pots attached to the top of jars containing *P. oligandrum* to facilitate the exposure of the roots to *P. oligandrum* VOCs. The result showed that the *N. benthamiana* plants exposed to Po-VOCs for 21 days grew noticeably more than the control plants, suggesting that Po-VOCs positively influenced the growth of *N. benthamiana* (Fig. 2A; Fig. S2). Moreover, considerable increases were observed in shoot length (1.4-fold) and dry weight (2.1-fold) of *N. benthamiana* plants that were exposed to Po-VOCs compared to the control (Fig. 2A). Additionally, the effects

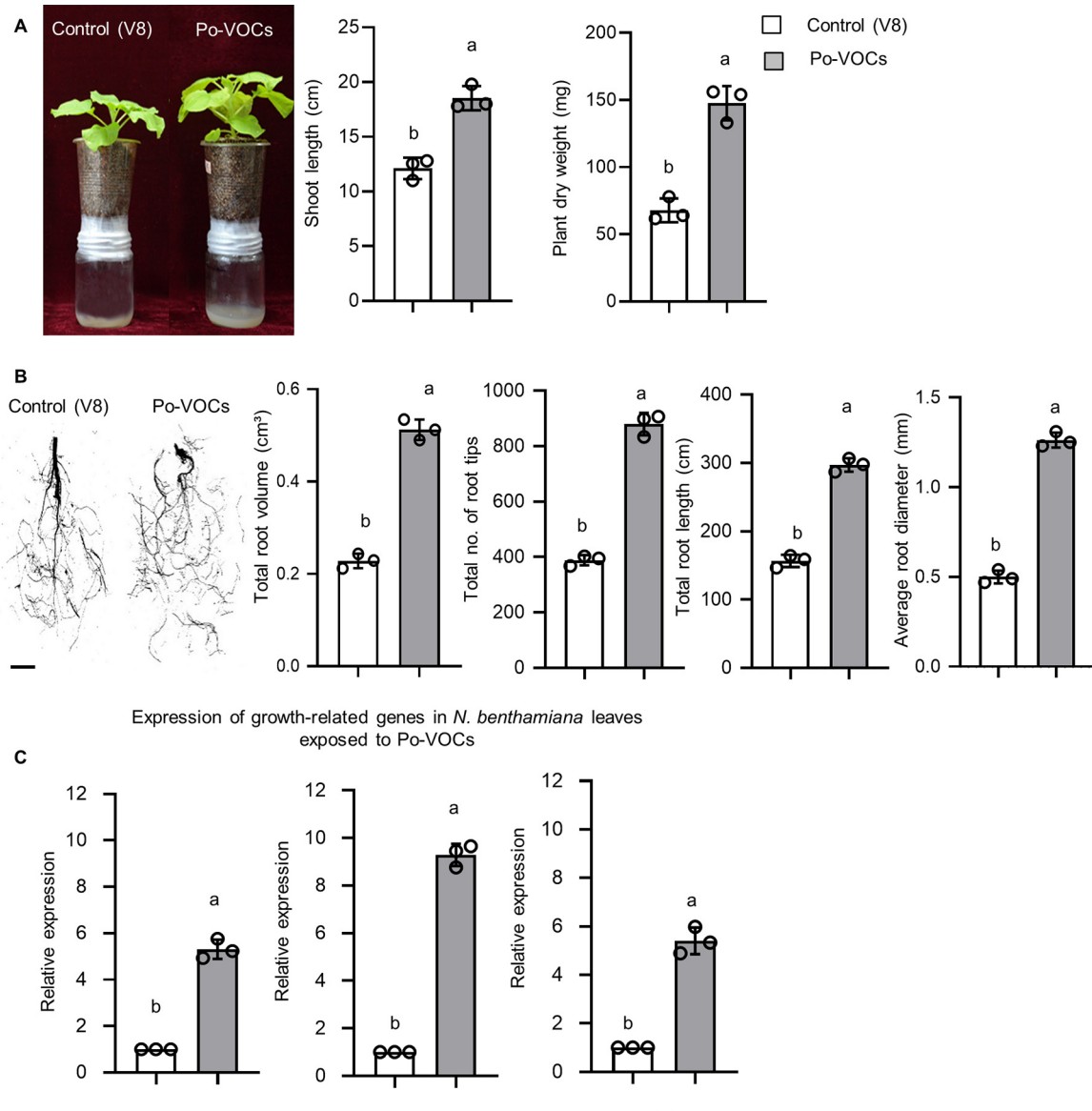

**FIG 2** Volatile organic compounds produced by *P. oligandrum* enhanced the growth of *Nicotiana benthamiana in planta*. (A) The plants were imaged 21 days after exposure to the control (V8) or Po-VOCs to see the difference in growth by measuring the shoot length and dry weight of the plants. (B) Representative images of the roots of *N. benthamiana* plants exposed to Po-VOCs or the control (V8) for 21 days. (The scale bar represents 2 cm.) Root morphological parameters, including total root volume, total number of root tips, total root length, and average root diameter, were measured. (C) Expression of the genes related to growth in leaves of *N. benthamiana* after 21 days of exposure of plants to the VOCs produced by *P. oligandrum*. qRT-PCR was performed using *NbEF-1a* as an internal reference. Error bars indicate the standard error ($n = 3$). Different lowercase letters above the bars represent significant differences between treatments. Experiments were repeated three times with similar results, and the results of second and third repeats of experiment are shown in Fig. S2.

of Po-VOCs on the root morphology were investigated. The root images of *N. benthamiana* taken after 21 days of Po-VOC exposure showed that there was an increase in root growth in plants that were exposed to Po-VOCs. The roots of plants that were exposed to Po-VOCs demonstrated significant increases in root volume (2.5-fold), number of root tips (2.2-fold), total root length (2.2-fold), and average root diameter (3.5-fold) compared to the control plants (Fig. 2B).

After 21 days of exposure to *P. oligandrum* VOCs, the expression levels of *N. benthamiana* genes related to growth were analyzed to see if the increased plant growth was associated with changes in gene expression. Our results showed that there was an increase in the expression of genes for expansin (5.2-fold in *N. benthamiana NbEXPA1* and 9.6-fold in *NbEXPA2*) and gibberellin-regulated protein (*NbGA14* [5.4-fold]) in *N. benthamiana* plants

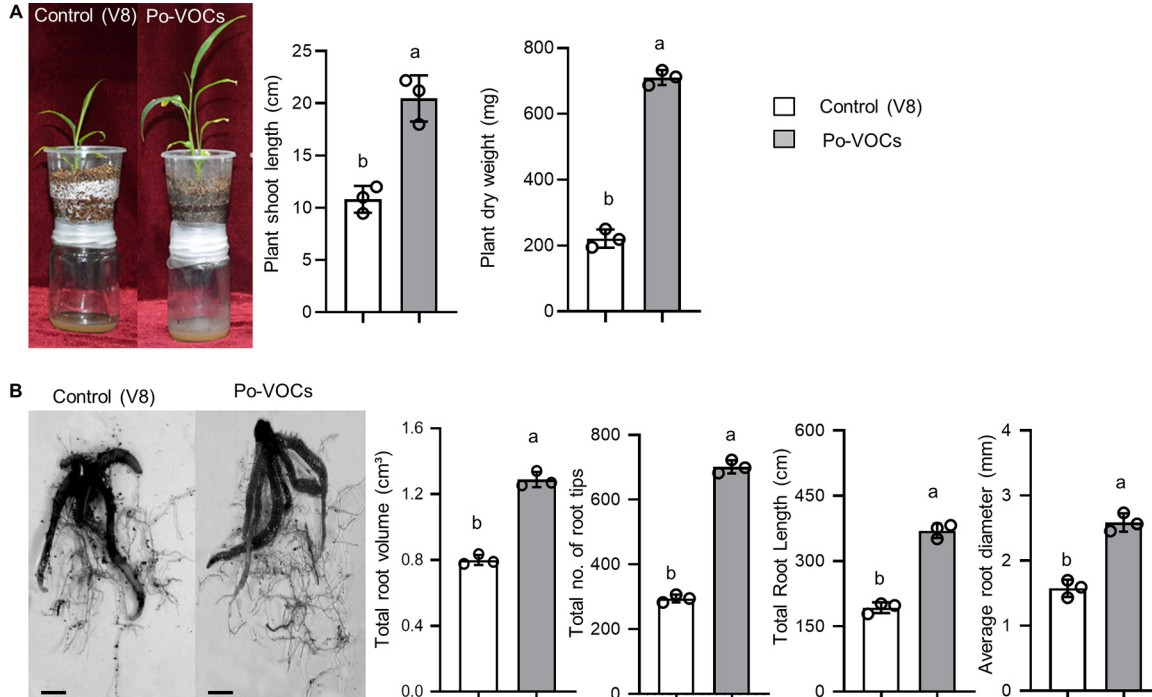

**FIG 3** Volatile organic compounds produced by *P. oligandrum* enhanced the growth of ginger (*Zingiber officinale*) plants. (A) The plants were imaged 21 days after exposure to Po-VOCs or the control (V8) to see the difference in growth by measuring the shoot length and dry weight of the plants. (B) Representative images of the roots of ginger plants exposed to Po-VOCs or the control (V8) for 21 days. (The scale bar represents 2 cm.) Root morphological parameters, including total root volume, the total number of root tips, total root length, and average root diameter, were measured. Error bars indicate the standard error (*n* = 3). Different lowercase letters above the bars represent significant differences between treatments. Experiments were repeated three times with similar results, and the results of the second and third repeats of the experiment are shown in Fig. S3.

exposed to Po-VOC compared to the control (Fig. 2C). The expansins play a role in cell expansion, elongation, and growth. *NbGA14* is regulated by a plant hormone, gibberellin, with higher expression in plants with enhanced growth. Overall, we found that *N. benthamiana* in the presence of Po-VOCs had a significant increase in the expression of growth-related genes and biomass content. After screening the potential of *P. oligandrum*-produced VOCs on *N. benthamiana* growth, we studied the growth-promoting potential of these VOCs on ginger.

**P. oligandrum VOCs enhanced the growth in ginger plants.** In an *in planta* assay, the growth-promoting potential of Po-VOCs on ginger plants was evaluated after 21 days. The results showed significant increases in shoot length (2-fold) and plant dry weight (3.3-fold) of Po-VOC-exposed ginger plants compared to the control (Fig. 3A; Fig. S3). Additionally, the root images of ginger taken after 21 days of Po-VOC exposure showed increased root growth. The root morphological parameters showed that there were significant increases in root volume (1.5-fold), number of root tips (2.3-fold), total root length (2.5-fold), and average root diameter (1.5-fold) in the plants that were exposed to the VOCs produced by *P. oligandrum* compared to the control plants (Fig. 3B). These results demonstrated that exposure to Po-VOCs increased the growth of ginger plants.

Interestingly, the trend regarding which of the rhizoscanner measurements had a bigger fold change (FC) increase differed between *N. benthamiana* and ginger. In *N. benthamiana*, the FC increases for root diameter (3.5-fold) and volume (2.5-fold) were larger than those for root length (2.2-fold) and number of root tips (2.2-fold) (Fig. 2). In ginger, the FC increases for root length (2.5-fold) and number of root tips (2.3-fold) were larger than those for root diameter (1.5-fold) and root volume (1.5-fold) (Fig. 3).

**P. oligandrum VOCs led to expression changes in ginger genes related to growth and defense.** To gain insight into the mechanism on how *P. oligandrum* VOCs lead to the enhancement of plant growth in ginger, transcriptome sequencing (RNA-seq) analysis was performed. Ginger leaves were sampled after 21 days of Po-VOC exposure and

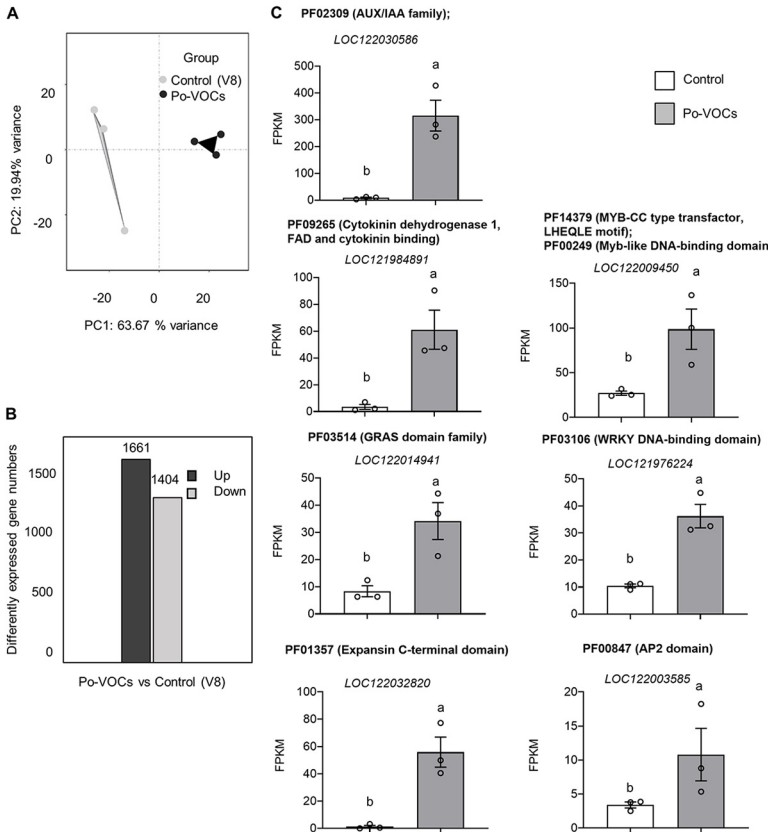

**FIG 4** (A) Principal-component analysis (PCA) of the six ginger (*Zingiber officinale*) transcriptome samples showing a clear separation between ginger plants either exposed to Po-VOCs or the control (V8). (B) Numbers of up- and downregulated genes ($P_{adj}$ <0.05; FC > 2; FPKM > 1); (C) expression levels from RNA-seq for selected genes (based on Pfam annotations) related to growth and stress responses in ginger plants exposed to either Po-VOCs or the control (V8). The error bars represent standard errors (*n* = 3). Different lowercase letters above the bars represent significant differences between treatments at $P_{adj}$ = 0.05 from the DESeq2 analysis.

compared to those of the ginger plants grown in the absence of Po-VOCs. Principal-component analysis (PCA) showed a clear separation between the three ginger plants that were exposed to Po-VOCs and the three control ginger plants grown without exposure to Po-VOCs (Fig. 4A). In terms of the numbers of differentially expressed genes (adjusted *P* value [$P_{adj}$], <0.05; fold change [FC], >2; fragments per kilobase per million [FPKM], >1), there were 1,661 genes upregulated and 1,404 genes were downregulated (Fig. 4B).

The transcriptomic data were analyzed for the functions of the differentially expressed genes (DEGs) based on Pfam domains that were used for functional annotation of ginger genes (see Table S2 in the supplemental material). The ginger genome contains at least 141 genes annotated with the Pfam domain for the auxin/indole-3-acetic acid (AUX/IAA) family (PF02309). The transcription of genes belonging to the AUX/IAA family can be upregulated by the plant hormone auxin (37). A substantial proportion (38/43) of DEGs annotated with this Pfam domain were upregulated with an average of 9-fold in the leaves of ginger plants exposed to Po-VOCs (Table S2B). Out of 21 genes annotated with a Pfam domain for cytokinin dehydrogenase 1 (CKX1), flavin adenine dinucleotide (FAD), and cytokinin binding domain (PF09265), all six of the DEGs were upregulated with an average of 8.8-fold change (Table S2B). The cytokinin binding domain binds cytokinin and starts a signaling cascade that regulates cell division and differentiation (38). Proteins with the GRAS (gibberellic acid insensitive [GAI], repressor of GA1-3 [RGA], and Scarecrow [SCR]) domain family (PF03514) can be involved in plant growth (39) and can be upregulated by the endogenous production of GA3 (40). There were at least 132 genes annotated with this domain family in the ginger genome, and most (19/22) of the DEGs were upregulated, with an average of

3.25-fold change in VOC-exposed ginger plants. There were at least 79 genes annotated with the expansin C-terminal domain (PF01357) in the ginger genome, and 13 out of 18 DEGs were upregulated, with an average of 17-fold in the Po-VOC-exposed ginger plants (Table S2B). Expansin can function to bind and loosen the cellulose microfibrils in the plant cell wall, allowing for the expansion and growth of the cell (41).

Transcriptomic data showed that in Po-VOC-exposed plants, there was an upregulation of genes that are potentially involved in regulating stress responses (Table S2B). The *Myb* and *WRKY* genes can be involved in the regulation of defense and stress responses (42, 43). In VOC-exposed ginger plants, out of at least 308 genes annotated with Pfam for the WRKY DNA-binding domain (PF03106), 37/53 DEGs were upregulated, with an average of 13-fold change in expression. The genome showed that there were at least 47 genes annotated with the Pfams for both the Myb-like DNA-binding domain (PF00249) and MYB-CC-type transfactor LHEQLE motif (PF14379). All of the DEGs were significantly upregulated (10/10), with an average of 6.4-fold in Po-VOC-exposed ginger plants compared to the control (Table S2B). The AP2 domain (PF00847) is a protein domain found in transcription factors important in regulating gene expression in plants in response to stress (44). The genome showed that there were at least 486 genes annotated with the Pfam domain for AP2 and 43 out of 57 DEGs annotated with this domain were significantly upregulated, with an average of 9.5-fold in the Po-VOC-exposed ginger plants (Table S2). The expression levels of representative genes from the above-described Pfam domains are shown in Fig. 4C. These representative genes were also selected for validation of the RNA-seq data set by quantitative PCR (qPCR) and are shown in Fig. S4, which showed the same trends in expression shown in Fig. 4C.

**The concentration of ginger growth hormones increased in the presence of *P. oligandrum* VOCs.** Growth hormones were selected for quantification in ginger leaf blades based on the upregulation of genes related to these hormones. Overall, our results showed that exposure to Po-VOCs increased the concentration of plant growth-related hormones in ginger. There was a significant increase in hormone concentration in leaf blades after 7, 14, and 21 days of Po-VOC-exposed ginger plants compared to the control. The concentration of indole-3-acetic acid (IAA) was increased significantly by 2.5-fold after 7 days and 3-fold after 14 and 21 days in ginger plants exposed to Po-VOCs compared to their controls (Fig. 5; Fig. S5). The concentration of zeatin (ZEA) increased significantly in Po-VOCs exposed ginger plants by 6-fold after 7, 14, and 21 days, whereas the concentration of gibberellic acid (GA3) increased by 4.3-fold after 7 days, 5.4-fold after 14 days, and 8-fold after 21 days of Po-VOC exposure (Fig. 5). The concentration of abscisic acid (ABA) was increased in Po-VOC-exposed ginger plants by 1.5-fold after 7 days and 1.7-fold after 14 days whereas at 21 days, the concentration was 3-fold lower than that of the control plants (Fig. 5). The lower concentration of ABA at 21 days was in contrast to the consistent increases in IAA, ZEA, and GA3 across all three time points in Po-VOC-exposed ginger plants.

**Individual VOCs 3-octanone and hexadecane identified by GC-MS enhanced *N. benthamiana* growth.** Twelve *P. oligandrum* GAQ1-produced VOCs (which had a relative percentage of area of >1% in the gas chromatagraphy-mass spectrometry [GC-MS] chromatogram) identified from our previous study (11) were tested individually for their plant growth-promoting ability by measuring seedling dry weight. Out of 12 VOCs, only two, hexadecane and 3-octanone, resulted in increased *N. benthamiana* seedling growth (Fig. S6). Although there was no increase in growth from the other 10 VOCs, they at least did not significantly reduce the seedling dry weight (Fig. S6). Both hexadecane and 3-octanone increased *N. benthamiana* seedling growth more at a concentration of 50 $\mu$M than at 10 $\mu$M, with maximum increases in dry weight of 2.7-fold for hexadecane and 3-fold for 3-octanone (Fig. 6; Fig. S7). Both hexadecane and 3-octanone were later analyzed at 100 $\mu$M, where for hexadecane, the dry weight of the seedlings was not significantly higher than the 50 $\mu$M concentration. However, at 100 $\mu$M concentration of 3-octanone, the dry weight of the seedlings was significantly lower than the 50 $\mu$M concentration (Fig. 6).

GC-MS analysis was conducted to identify whether the two growth-promoting VOCs hexadecane and 3-octanone produced by *P. oligandrum* GAQ1 were also produced by *P. oligandrum* strain CBS 530.74. Our results showed that the *P. oligandrum* strain

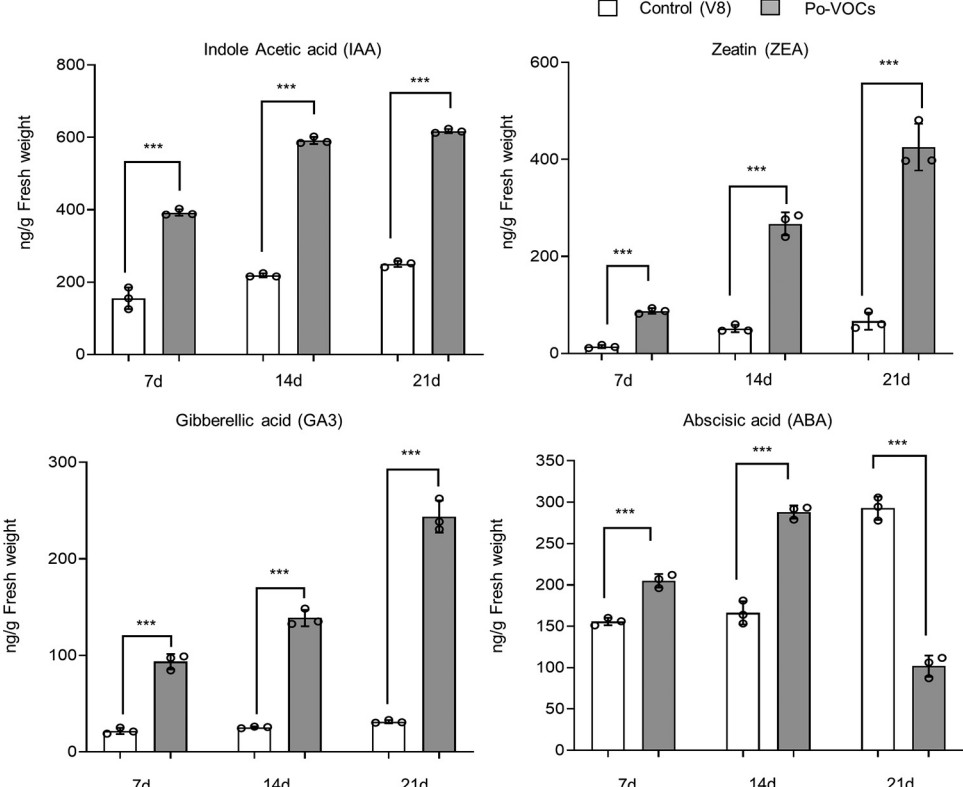

**FIG 5** Concentrations of selected plant growth hormones in leaf blades of ginger plants exposed to *P. oligandrum* volatile organic compounds for 7, 14, and 21 days. Error bars indicate the standard error ($n = 3$). ***, significant difference between treatment and control at the level of $P < 0.001$. The experiment was repeated twice with the same trends, and the results from the second repeat of the experiment are shown in Fig. S5.

CBS 530.74 produced hexadecane, whereas 3-octanone was not detected (Table S3 and Fig. S8). Moreover, the GC-MS chromatograph of *P. myriotylum* SWQ7 showed that both hexadecane and 3-octanone were not detected and standards were also used to confirm the identity of both VOCs (Fig. S8).

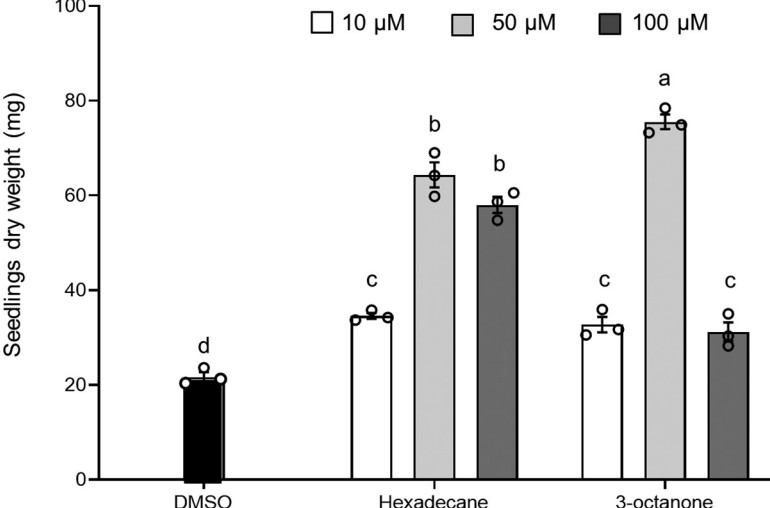

**FIG 6** Effect of two growth-promoting VOCs at a higher concentration (100 $\mu$M) on *N. benthamiana* seedlings' growth. *N. benthamiana* seedlings were exposed to either individual VOCs, hexadecane and 3-octanone, or the control (DMSO) for 10 days, and the dry weight of the seedlings was recorded. Error bars indicate the standard error of the mean ($n = 3$). Lowercase letters above the bars indicate a significant difference between treatments. Experiments were replicated at least three times, producing similar results, and the results from the second and third repeats of the experiment are shown in Fig. S7.

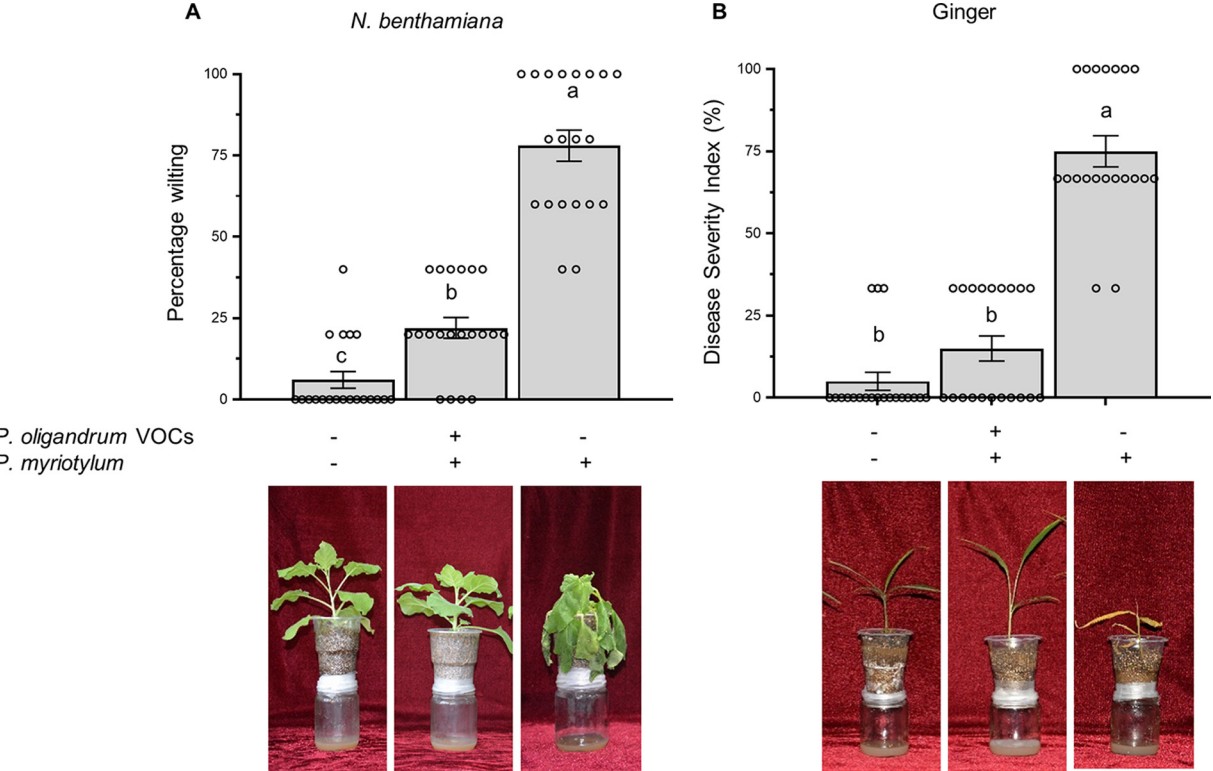

**FIG 7** *P. oligandrum* VOCs reduced the disease severity in *N. benthamiana* and ginger plants. (A) *P. myriotylum*-infected *N. benthamiana* plants were exposed to either the control (V8) or Po-VOCs for 21 days, whereas the noninoculated *N. benthamiana* plants without exposure to Po-VOCs were used as negative controls. The plants were photographed, and the symptoms were scored after 21 days of inoculation of *P. myriotylum* and exposure to Po-VOCs to calculate the percentage of wilting in *N. benthamiana*. (B) *P. myriotylum*-infected ginger plants were exposed to either Po-VOCs or the control (V8) for 21 days, whereas the noninoculated ginger plants without exposure to Po-VOCs were used as negative controls. The plants were photographed, and the symptoms were scored after 21 days of inoculation of *P. myriotylum* and exposure to Po-VOCs to calculate the disease severity index in ginger. Error bars are the standard error of the mean ($n = 20$). Lowercase letters on the bars show significant differences between different treatments. The experiment was repeated twice with similar results, and the results from the second repeat of the experiment are shown in Fig. S9.

**P. oligandrum VOCs reduced disease symptoms in ginger plants.** To demonstrate the effect of *P. oligandrum*-produced VOCs on the disease severity in ginger, the plants infected with the soft rot pathogen *P. myriotylum* were exposed to *P. oligandrum* VOCs. After 21 days of exposure to Po-VOCs, the plants were photographed, and the disease severity index was calculated. Our results showed that 75% disease severity was observed in the *P. myriotylum*-infected ginger plants without VOC exposure, while those exposed to *P. oligandrum* VOCs had a much lower disease severity index of 15% (Fig. 7; Fig. S9). *P. myriotylum*-infected *N. benthamiana* plants were also evaluated for the development of disease in the presence and absence of VOCs produced by *P. oligandrum*. *P. myriotylum*-infected *N. benthamiana* plants without VOC exposure showed wilting symptoms, with a wilt index of 78%; however, the wilt index was reduced to low as 22% in plants that were exposed to Po-VOCs (Fig. 7). These results demonstrated that Po-VOCs could reduce the disease severity in ginger and *N. benthamiana* plants infected with *P. myriotylum*.

## DISCUSSION

*P. oligandrum* has been reported to promote growth (7, 45) and induce systemic resistance in plants (46–48). Previous studies have demonstrated that *P. oligandrum* promotes plant growth by producing tryptamine (TNH$_2$), an auxin-like compound (7, 49), and induces systemic resistance by microbe-associated molecular patterns (8, 50, 51). However, the role of *P. oligandrum* VOCs in plant growth promotion and induction of disease resistance has not yet been studied. Here, we show that the production of VOCs by *P. oligandrum* could be another factor in *P. oligandrum*-mediated plant growth promotion and disease resistance.

In the present study, we observed an increase in the growth of *N. benthamiana* seedlings

and plants exposed to Po-VOCs. VOCs produced by *B. subtilis* (1), *Pseudomonas fluorescens* (52), and *Trichoderma* spp. (53) have been reported to enhance plant growth. In *N. benthamiana* plants exposed to Po-VOCs, the increased growth was positively correlated with the significant upregulation in the expression of the expansin genes *NbEXPA1* and *NbEXPA2* (Fig. 2C). *B. subtilis* VOCs have been shown to upregulate the expression of expansin genes in *Nicotiana tabacum* (1). Previous studies showed that the expression level of *NbEXPA1* was significantly higher in *N. benthamiana* plants with enhanced growth (54). Exposure to Po-VOCs resulted in significantly higher expression of a gibberellin-regulated protein encoded by *NbGA14*. In *N. benthamiana* mutants where genes from the GA signaling pathway were deleted, the relative expression level of *NbGA14* was low, implicating *NbGA14* as GA responsive (55). Our results from *N. benthamiana* are in line with how exposure to VOCs produced by *M. aurantiacum* enhanced the growth of *N. benthamiana* through auxin and gibberellin pathways (24).

Microbial VOCs have been shown to increase growth and induce systemic resistance in plants, including *A. thaliana* (56, 57) and tobacco (24). Here, we used ginger after screening the potential of Po-VOCs for *N. benthamiana* growth enhancement and demonstrated that growth was also increased in Po-VOC-exposed ginger plants. Transcriptomic analysis of the ginger leaves suggested that exposure to Po-VOCs modulates hormone signal transduction in ginger involving auxins, cytokinins, and gibberellins. A substantial number of genes annotated with the Pfam domain for the AUX/IAA family (PF02309) were upregulated in Po-VOC-exposed ginger plants. The plant hormone auxin can trigger the transcription of genes belonging to the AUX/IAA family (58). Auxin plays a critical role in the regulation of various aspects of plant growth and development, particularly root growth (59). The enhanced growth in Po-VOC-exposed ginger plants can positively be correlated with the increased concentration of the auxin IAA. Zeatin is a plant cytokinin that is primarily synthesized in the roots and transported to other parts of the plant, where it can promote shoot and leaf growth (60). In Po-VOC-exposed ginger plants, an increased concentration of zeatin was positively correlated with the enhanced shoot and root growth. In Po-VOC-exposed ginger, there was upregulation of the genes that were annotated with a Pfam domain for cytokinin dehydrogenase 1, FAD, and cytokinin binding (PF09265), which could potentially be involved in maintaining cytokinin homeostasis in Po-VOC-exposed ginger plants. Cytokinin dehydrogenase genes can regulate cytokinin homeostasis to maintain balanced hormone levels such as in rice (61). Gibberellin (GA) signaling controls a variety of plant growth and development characteristics and is primarily regulated by proteins in the GRAS domain family (PF03514) (62, 63). In Po-VOC-exposed ginger plants, there was significant upregulation of genes annotated with a Pfam domain for the GRAS family, which could potentially be involved in GA signaling in Po-VOC-exposed ginger plants. In addition, the increased concentration of endogenous GA3 in Po-VOC-exposed ginger plants could positively be correlated with increased growth as it has been reported that endogenous GA3 in *Arabidopsis* upregulates the expression of genes in the GRAS domain family (40). Previously it has been demonstrated that spraying GA increased the shoot length of ginger and weight of ginger rhizome (64). Similarly, the yield of ginger rhizomes was significantly increased by the application of GA3 (65, 66). In Po-VOC-exposed ginger plants, there was an upregulation in the expression of genes annotated with the Pfam domain for expansin C-terminal domain (PF01357), which can positively correlate with the growth enhancement in Po-VOC-exposed ginger plants as expansin proteins are involved in cell expansion and elongation (67).

Transcriptomic analysis showed that exposure to Po-VOCs upregulated genes related to various stress responses. A significant proportion of the putative genes annotated with the Pfam domains for the WRKY DNA-binding domain (PF03106), Myb-like DNA-binding domain (PF00249) and MYB-CC type transfactor LHEQLE motif (PF14379), and AP2 domain (PF00847) were upregulated. WRKY transcription factors can be important for the regulation of genes associated with plant defense responses. Our results showed that genes annotated with the Pfam domain for the WRKY DNA-binding domain were upregulated in Po-VOC-exposed ginger plants. Previously, the overexpression of *Vitis vinifera* WRKY2 (*VvWRKY2*) in

tobacco was shown to enhance resistance against *Pythium* spp. (68). It has been shown that *Zingiber officinale* WRKY8 (*ZoWRKY8*) was upregulated in ginger plants infected with *Fusarium oxysporum* (69). In ginger exposed to Po-VOCs, the upregulation of genes annotated with the Pfam database for both the Myb-like DNA-binding domain (PF00249) and MYB-CC type transfactor LHEQLE motif (PF14379) suggested that exposure to Po-VOCs could be related to stress responses. Previously, it has been reported that *ZoMYB* transcription factors were upregulated in abiotically stressed ginger (70). In Po-VOC-exposed ginger plants, the upregulated genes annotated with the Pfam database for AP2 (PF00847) could be involved in priming ginger plants for resistance to abiotic stresses. Previously, ginger genes annotated with the AP2 domain were upregulated under abiotic stresses (71).

Individual VOCs produced by *P. oligandrum*, reported in our previous study (11), were tested for plant growth promotion. Out of 12 VOCs, two VOCs, hexadecane and 3-octanone, significantly enhanced the growth of *N. benthamiana* seedlings. Previously, hexadecane has been shown to increase *Arabidopsis* growth (27). Interestingly, the growth promotion by 3-octanone was less at a higher concentration, where at 100 $\mu$M, the dry weight was significantly lower than at 50 $\mu$M (Fig. 6). It has previously been demonstrated that much higher 3-octanone ($\sim$5 mM) concentrations than those used in our study decreased growth in *Arabidopsis* (72).

There was a reduction in disease severity in *P. myriotylum*-infected ginger plants that were exposed to Po-VOCs compared to nonexposed ginger plants. In our previous study, we demonstrated that VOCs produced by *P. oligandrum* inhibited the growth of *P. myriotylum* (11), which could be one of the potential contributions of Po-VOCs to disease control in ginger. Previous studies showed that VOCs produced by biocontrol *Bacillus* spp. resulted in disease control via inhibition of plant-pathogenic *Sclerotinia sclerotiorum* (73). Similarly, the exposure of tobacco plants to VOCs produced by *B. subtilis* enhanced the resistance against bacterial wilt disease and reduced disease severity via inhibition of *Ralstonia solanacearum* (74). In our study, GC-MS analysis showed that the two growth-promoting VOCs (3-octanone and hexadecane) could not be detected from *P. myriotylum* growing on V8 medium, indicating that the two VOCs are probably produced by only *P. oligandrum* in the disease biocontrol experiment. Regarding biocontrol applications, our results demonstrated that Po-VOCs enhanced growth and reduced disease severity in ginger plants. Commercial *P. oligandrum* products, such as Polyversum, are already available and recommended for use in various crops for disease control (75). Based on our finding that *P. oligandrum* produces plant growth-promoting VOCs, the application of *P. oligandrum* biocontrol products could be enhanced. However, commercial strains should be tested to determine whether they produce growth-promoting VOCs, or the strain used in our study can be used to develop a commercial product.

In conclusion, the data presented in our study demonstrated that VOCs produced by *P. oligandrum* promoted the growth of *N. benthamiana* and ginger. Transcriptomic analysis showed that exposure to Po-VOCs induced the differential expression of putative ginger genes for growth promotion, resistance, and hormone signaling. The increased hormone concentrations in the leaves of ginger further suggested the involvement of hormones in signaling pathways important for VOC-mediated growth promotion and reduction of disease severity in ginger. Overall, our findings provide new insights into the potential of *P. oligandrum* in VOC-mediated plant growth promotion and induction of disease resistance in plants.

## MATERIALS AND METHODS

**Pythium strains, plant material, and growth conditions.** The *P. oligandrum* strain GAQ1 (CGMCC no. 17470) was described previously for its biocontrol activity (5, 76). The *P. myriotylum* strain SWQ7 CGMCC no. 21459 was described previously (13) as a pathogen causing *Pythium* soft rot disease of ginger. The biocontrol *P. oligandrum* strain CBS 530.74 was described previously (77). All of the strains were grown on 10% V8 agar medium at 25°C. Ginger plants ("Laiwu" variety) were derived from tissue culture, transplanted to autoclaved vermiculite in 100-mL pots, and grown in 16-h-light and 8-h-dark cycles at 25°C in a growth chamber. *N. benthamiana* plants were grown from seeds that were surface sterilized by soaking in 70% ethanol for 1 min, followed by soaking in 1% sodium hypochlorite for 15 min, rinsed four to five times in sterile distilled water, and dried on a filter paper. The seeds were then placed in a growth chamber in 16-h-light and 8-h-dark

cycles at 25°C. Young seedlings were then transplanted into plastic pots (one seedling per pot) and grown under the growth conditions described above.

**Effect of *P. oligandrum* VOCs on *N. benthamiana* seedling growth *in vitro*.** To investigate the effect of VOCs produced by *P. oligandrum* on seedling growth, five *N. benthamiana* seedlings of equal size were placed on one side of a bipartite petri plate containing Murashige and Skoog (MS) medium (0.8% agar, 1.5% sucrose [pH 5.7]), a 5-mm mycelial plug of actively growing *P. oligandrum* was placed on the other side of the petri plate containing V8 agar medium, and the plates without *P. oligandrum* were used as a control. The petri plates were sealed by wrapping them at least nine times with Parafilm (Parafilm M; Bemis, USA) to avoid the escape of VOCs and placed in an environmentally controlled chamber with 16-h-light and 8-h-dark cycles at 25°C. The seedling shoot length and dry weight were measured after 10 days. To measure the dry weight, plant material was dried in an oven until it reached a constant weight, which indicated that all moisture was removed. The experiment was repeated three times, each time with three replicates, and each replicate included five seedlings. Root morphological parameters were measured using a RhizoScan scanner (Epson Perfection V700 Photo; Epson, USA) and analyzed by WinRHIZO software (Regent Instruments Co., Canada) (78).

**Effect of *P. oligandrum* VOCs on plant growth in *N. benthamiana* and ginger.** The effect of *P. oligandrum* GAQ1 VOCs (Po-VOCs) on *N. benthamiana* and ginger *in planta* was evaluated by the method described previously (52), and a diagram illustrating the pot-jar assembly used in our study is shown in Fig. S10 in the supplemental material. Glass jars containing V8 agar medium were inoculated with a 5-mm actively growing mycelial plug of *P. oligandrum*, whereas the noninoculated glass jars containing only V8 medium were used as a control. Six small 2-mm holes were made at the bottom of the plastic pots (270 mL) to allow the roots to be exposed to the VOCs produced by *P. oligandrum*. Two sterilized filter paper pieces (catalog no. 99-102-150; Cytiva, USA) were placed inside the bottom of the plastic pot to avoid the leakage of contaminating liquid through the holes into the glass jar. Equal-sized *N. benthamiana* or ginger seedlings (one in each pot) were transferred to the plastic pots containing sterilized vermiculite and organic matter (3:1) and placed on the glass jars containing either *P. oligandrum* or only V8 agar medium as control. The pot-jar assembly was then firmly sealed by wrapping it at least nine times with parafilm (Parafilm M; Bemis, USA) to avoid the escape of VOCs produced by *P. oligandrum*. The pot-jar assembly was then subjected to 16-h-light and 8-h-dark cycles at 25°C and watered with sprayer. Plants were watered by wetting the surface of the soil to avoid contamination by leakage of water through holes into the glass jar. Plants were grown for 21 days, and growth was evaluated by the difference in shoot lengths (from the base of the stem to the top of the straightened leaves), dry weights of the whole plant, and root morphological parameters. Leaf material was collected and frozen at −80°C for subsequent analysis of gene expression and hormone levels. For the gene expression and hormone analyses, separate experiments were performed. Three biological replicate leaf samples were collected for each treatment: i.e., three biological replicate samples for VOC-exposed plants and three biological replicate samples for nonexposed (control) plants.

**Experimental design, RNA extraction, RNA-seq, and analysis.** For transcriptome analysis, the ginger plants were grown as described in the previous section. The leaf blades from the two uppermost leaves of the plants grown for 21 days with and without exposure to Po-VOCs were collected from a separate experiment from that used for hormone analysis, flash-frozen in liquid nitrogen, and stored at −80°C. For the RNA sequencing analysis, three biological replicate samples were collected for VOC-exposed and nonexposed (control) plants. Total RNA was extracted using the TRIzol reagent according to the manufacturer's protocol. RNA purity and quantity were evaluated using the NanoDrop 2000 spectrophotometer (Thermo Scientific, USA), and the parameter to check the quality of RNA samples included measuring the absorbances of $A_{260/280}$ and $A_{260/230}$, whereby for all the samples, $A_{260/280}$ was greater than 2.0 and $A_{260/230}$ was greater than 1.8, and RNA integrity was evaluated using the Agilent 2100 bioanalyzer (Agilent Technologies, USA). The libraries were constructed using the TruSeq stranded mRNA LT sample prep kit (Illumina, USA) according to the manufacturer's instructions.

The libraries were sequenced on an Illumina HiSeq X 10 platform, and 150-bp paired-end reads were generated. Raw data (raw reads) of fastq format were first processed using Trimmomatic (79), and the low-quality reads were removed to obtain the clean reads. The clean reads were mapped to the *Zingiber officinale* reference genome (GCF_018446385.1_Zo_v1.1_genomic.fna) (80) using HISAT2 (81), and >90% of the reads were mapped to the genome from all of the samples. The number of fragments per kilobase of gene model per million mapped reads (FPKM) (82) of each gene was calculated using Cufflinks (83), and the read counts of each gene were obtained by HTSeq-count (84), where the GFF file GCF_018446385.1_Zo_v1.1_genomic.gff was used for the gene annotations. Differential expression analysis was performed using DESeq2 (85). An adjusted *P* value ($P_{adj}$) of <0.05, fold change (FC) of >2 or <0.5, and an FPKM of >1 in the up- or downregulated genes were set as the threshold for significantly differential expression. The PCA was performed using the R statistical environment. The ginger proteins were annotated with Pfam domains using the InterProScan functional annotation tool (Galaxy version 5.55–88.0+galaxy3) using the default settings at https://usegalaxy.eu (86–88). The ginger protein sequences were downloaded from NCBI using the GCF_018446385.1_Zo_v1.1_protein.fasta file from reference 80.

**Real-time PCR analysis of *N. benthamiana* and ginger gene expression.** The expression in *N. benthamiana* leaves of genes annotated as encoding expansins (*NbEXPA1* and *NbEXPA2*) and gibberellin-regulated protein (*NbGA14*) was measured. Expansin genes *NbEXPA1* and *NbEXPA2* were amplified using primers from references 1 and 54, and the *NbGA14* gene was amplified using primers from reference 55. The details of the primers are listed in Table S1 in the supplemental material. *N. benthamiana* leaves (excluding the petiole) were harvested from plants after 21 days of exposure to Po-VOCs, and the plants without VOC exposure served as the control. The qPCR validation for the ginger RNA-seq was performed using the same RNA samples used for RNA-seq. cDNA was synthesized using the EasyScript one-step genomic DNA (gDNA) removal and cDNA

synthesis kit (TransGen Biotech), using random primers according to the manufacturer's instructions. The qPCR was performed with a LightCycler 96 instrument (Roche Life Sciences, USA) using SYBR green Pro *Taq* HS premix (Accurate Biotechnology, China) with *NbEF-1α* (89) and *ZoEF-1α* (90) as an internal reference. The quantitative reverse transcription-PCR (qRT-PCR) program consisted of denaturation at 95℃ for 3 min, followed by 40 amplification cycles at 95℃ for 10 s and 60℃ for 30 s. A melt curve analysis step was included to confirm the specificity of the primer pairs used. Each sample was replicated three times for qPCR, the data were analyzed using the LightCycler 96 software, and the relative expression was calculated by the threshold cycle ($2^{-\triangle\triangle CT}$) method (91). For ginger, the primers were designed using Primer-BLAST software (92) at NCBI and are listed in Table S1.

**Quantification of growth hormones in ginger plants exposed to *P. oligandrum* VOCs.** Ginger plants exposed to Po-VOCs were analyzed for the concentrations of the plant growth-regulating hormones gibberellic acid (GA3), indole 3-acetic acid (IAA), zeatin (ZEA), and abscisic acid (ABA). Leaf blades were collected at three time points of 7, 14, and 21 days from both Po-VOC-exposed and nonexposed ginger plants (grown as described in the previous section). For the extraction of the hormones, 0.1 g of fresh (not dried) leaf blades from each treatment was ground to a fine powder using a TissueLyser (60 s at 50 Hz). The fine leaf powder was then used to prepare the samples for hormone quantification based on a method described previously (93). Briefly, the powder was extracted with 1 mL of 80% methanol in a 2-mL tube by shaking at 300 rpm overnight at 4℃. The extracted samples were vortexed and centrifuged at 14,000 rpm at 4℃ for 30 min. The supernatant from the samples was transferred to a new tube and then dried by vacuum evaporation at room temperature for 6 h, and then 200 $\mu$L of 80% methanol was added to the dried samples to dissolve the extract and then passed through a 0.2-$\mu$m-pore-size filter. The standard chemicals were purchased from Macklin, China for each of the quantified hormones (IAA, GA3, ZEA, and ABA) and standard solutions of different concentrations (0.1, 0.2, 0.5, 2, 5, 20, 50, and 200 ng/mL) were prepared in 80% methanol as described previously (93), and samples were run on a Xevo TQ-S (Waters, USA) machine coupled with the Waters Acquity HSS T3 ultraperformance (UPLC) column (1.8 $\mu$m, 2.1 mm by 75 mm) with an injection volume of 2 $\mu$L at a column temperature of 30℃ for detection and quantification (94).

**Effect of individual VOCs on *N. benthamiana* seedling growth.** The *P. oligandrum* VOCs identified through previous GC-MS analysis based on their relative percentage of area of >1% (11) were evaluated using the individual compounds purchased from Macklin (China) or Aladdin (China) for their *N. benthamiana* seedling growth-promoting potential. The 12 compounds were methyl heptenone, 3-octanone, ethyl decanoate, 1-octanal, $\alpha$-pinene, D-limonene, dodecane, 2-undecanone, 2-heptanone, pinanediol, hexadecane, and 2-phenylethanol. Dimethyl sulfoxide (DMSO) was used as a solvent to obtain final concentrations of 10, 50, and 100 $\mu$M. *N. benthamiana* seedlings were grown on MS agar medium on one side of the bipartite plate as described in the previous section, and on the other side of the plate, 50 $\mu$L of each of the VOCs was applied to 1-cm-diameter filter papers. Control plates were prepared in the same manner, except the filter papers were wetted with only DMSO. After 10 days of VOC exposure, the dry weight of the seedlings was measured.

**GC-MS analysis of the VOCs produced by *P. oligandrum* and *P. myriotylum*.** Solid-phase microextraction (SPME) gas chromatography-mass spectrometry (GC-MS) analysis was conducted to identify VOCs produced by *P. oligandrum* strains GAQ1 and CBS 530.74 and *P. myriotylum* strain SWQ7. Five-millimeter-diameter mycelial plugs from an actively growing mycelial mat of either *P. oligandrum* (GAQ1, CBS 530.74) or *P. myriotylum* SWQ7 was inoculated on V8 agar medium in 100-mL vials. Noninoculated V8 agar medium was used as a control. The absence of a VOC from the noninoculated V8 agar medium control was critical to assigning a VOC as actually produced by *P. oligandrum* or *P. myriotylum*. The vials were closed with rubber lids and were firmly sealed by wrapping them at least nine times with parafilm (Parafilm M; Bemis, USA), and then the vials were incubated at 25℃ for 3 days before VOC collection. The SPME fiber was injected into the headspace of the vial containing either *P. oligandrum* GAQ1 or CBS 530.74 or *P. myriotylum* SWQ7 and incubated at 30℃ for 30 min. GC-MS analysis was carried out using a TSQ 8000 EVO gas chromatograph (Thermo Scientific, USA), and VOCs were identified as described in reference 95. Standards for hexadecane and 3-octanone (Macklin, China) were also run on the GC-MS machine.

**Effect of *P. oligandrum* VOCs on control of *P. myriotylum*-caused disease on ginger and *N. benthamiana*.** The 5-mm freshly growing mycelial plug of *P. oligandrum* GAQ1 was inoculated in the glass jars containing V8 agar medium, while the glass jars containing only V8 agar medium were used as the control. The plastic pots were prepared as described in the previous section. The roots were inoculated with autoclaved wheat seeds, which were used to culture *P. myriotylum* SWQ7 isolate or controls at the same time as transplantation of equal-sized seedlings of either *N. benthamiana* or ginger. The wheat seed inoculum was prepared by the method described previously (5). The plastic pots inoculated with *P. myriotylum*, and noninoculated controls were then fixed onto glass jars containing *P. oligandrum* and other glass jars containing only V8. The glass jars were then sealed by wrapping them at least nine times with Parafilm to avoid the escape of VOCs. The trial was carried out in a growth chamber at 16-h-light and 8-h-dark cycles at 25℃. Plants were observed regularly for the development of disease symptoms. The experiment was performed using a completely randomized design with 20 replicate plants in each experiment and repeated three times. The disease symptoms were scored 21 days after *P. myriotylum* inoculation using the formula disease severity index (%) = [$\Sigma$ ($ni \times vi$)/($V \times N$)] × 100, where *ni* indicates the number of plants with the respective disease rating, *vi* is the disease rating, *V* is the highest disease rating (3), and *N* is the number of plants observed. The ginger disease rating was calculated using the following scale described in reference 32: 0 = healthy plants; 1 = discolored sheath collar and yellow lower leaves; 2 = plants alive, but with some shoots either totally yellow or dead; and 3 = all shoots dead. The *N. benthamiana* wilting rating was calculated using the following scale: 1 = no symptoms, 2 = one leaf wilted, 3 = two to three leaves wilted, 4 = four or more leaves wilted, and 5 = whole plant wilted.

**Statistical analysis.** Generally, the data were statistically analyzed using either Student's *t* test or one-way analysis of variance (ANOVA) using SPSS software version 25, and the significant difference between the means was determined using Tukey's honestly significant difference (HSD) test after one-way ANOVA.

**Data availability.** The RNA-seq reads from this project were submitted to the GEO database under accession no. GSE235182.

## SUPPLEMENTAL MATERIAL

Supplemental material is available online only.

**SUPPLEMENTAL FILE 1**, XLSX file, 0.01 MB.
**SUPPLEMENTAL FILE 2**, XLSX file, 7.5 MB.
**SUPPLEMENTAL FILE 3**, PDF file, 1.8 MB.

## ACKNOWLEDGMENTS

This work was financially supported by the China Agriculture Research System of MOF and MARA (CARS-24-C-01), the Jiangsu Funding Program for Excellent Postdoctoral Talent (2022ZB769), the Jiangsu Agricultural Science and Technology Innovation Fund [CX (18)2005] and the Jiangsu Coast Land Resources Development Co., Ltd's 2022 Saline-alkali Land Management and Soil Fertility Enhancement Science and Technology "Open Competition Mechanism to Select the Best Candidates" Project (2022YHTDJB03). Assistance from OE Biotech Co. for next-generation sequencing and support staff from the JAAS central testing laboratory for GC-MS analysis are acknowledged.

We acknowledge Daolong Dou for the gift of the *P. oligandrum* CBS 530.74 strain.

L.W. and D.Z. are coauthors on a patent application for *P. oligandrum* (patent application no. CN201910757035.2) relating to the biocontrol of plant diseases using the *P. oligandrum* GAQ1 isolate.

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
