## [Reviewer comments · Microbiology Spectrum]

Microbiology Spectrum

Volatile organic compounds emitted by the biocontrol agent *Pythium oligandrum* contribute to ginger plant growth and disease resistance

Taha Majid Mahmood Sheikh, Dongmei Zhou, Haider Ali, Sarfraz Hussain, Nan Wang, Siqiao Chen, Yishen Zhao, Xian Wen, Xiaoyu Wang, Jinfeng Zhang, Lunji Wang, Sheng Deng, Hui Feng, Waseem Raza, Lihui Wei, and Paul Daly

Corresponding Author(s): Paul Daly, Jiangsu Academy of Agricultural Sciences

Review Timeline:

Submission Date:	April 13, 2023
Editorial Decision:	May 28, 2023
Revision Received:	June 23, 2023
Accepted:	June 26, 2023

Editor: Lindsey Burbank

Reviewer(s): The reviewers have opted to remain anonymous.

Transaction Report:

DOI: <https://doi.org/10.1128/spectrum.01510-23>

May 28, 2023

Dr. Paul Daly
Jiangsu Academy of Agricultural Sciences
Institute of Plant Protection
No. 50 Zhongling Street
Nanjing 210014
China

Re: Spectrum01510-23 (Volatile organic compounds emitted by the biocontrol agent *Pythium oligandrum* contribute to ginger plant growth and disease resistance)

Dear Dr. Paul Daly:

Thank you for submitting your manuscript to Microbiology Spectrum. The reviewers and myself appreciate the investigation of biological control interactions involving VOCs. Your manuscript is well written and shows some interesting results. There are just a few areas where added clarification would improve the transparency and replicability of your study. Please carefully address the reviewers comments as outlined below.

Link Not Available

Sincerely,

Lindsey Burbank

Journals Department
Reviewer comments:

Reviewer #2 (Comments for the Author):

In the present study, Sheikh et al showed that the volatile organic compounds produced by a biocontrol agent, *Pythium oligandrum* have the potential to enhance growth and induce resistance against various stress in plants. The authors showed

that VOCs produced by *P. oligandrum* also increased the expression of genes for growth-related hormones and various stress responses. From the analysis of the transcriptomic dataset, the authors quantified the growth-related hormones which were in higher concentration in the plants that were exposed to the VOCs, and it supported the growth enhancement in the presence of VOCs. The authors also showed that exposure to the VOCs also reduced the severity of the disease. Overall, the study is comprehensive and well-planned, introduction section is well written, methodology is good, and results are interesting. However, there are some suggestions that should be considered.

Abstract; Line 1; It is better that author write about 1-2 introductory sentences related to the importance of pathogen or disease.

Line 35; in sustainable agriculture by enhancing....

Line 36-37; is only the effect of VOCs was study related to plant growth not disease suppression? Make it clear write about disease suppression.

Line 37-39; why the tobacco plants were used? As the whole explained about the effect of VOCs on the ginger plant growth and disease severity.

Lines 45-47; it is better to mention the priming of ginger plants against various stress.

Line 45; Write about the conclusion of your findings.

Line 51-52; revise the sentence meanings are not clear, write such as "..... produced by *P. oligandrum* related to plant growth promotion and disease resistance".

Line 66; delete a non-plant pathogenic "an oomycete"

Line 69; delete "or by", and induction of host

Line 73; delete "recently"

Line 75; as author mentioned *Pythium myriotylum* having a broad host range, so its better to 2-3 examples of diseases caused by *Pythium myriotylum*

Line 89; author need to check in the whole manuscript, abbreviate the genus name when it appears 2nd time.

Line 90-91; revise the sentence

In lines 95-99, The sentence structure is not clear, need to revise

Line 101-104; same as above, need to abbreviate the genus names.

From lines 94-107, in the second last paragraph of the introduction, add some information about the biocontrol oomycete species.

In lines 108-114, some points from the first paragraph of the discussion (from lines 433-439) should go into the last paragraph of the introduction.

Line 120; *P. oligandrum* strain CBS is a plant pathogenic or biocontrol agent?

Line 121; for how long time the strain was grown?

In line 131, it is more useful to be consistent with mentioning the number of plants or seedlings used. Either use it every time in methodology or don't use it.

Line 132; delete "(grown as described in previous section)"

Line 139-140; number of seedlings per replications?

Line 144-145; rewrite the sentence

In line 153, if it is vermiculite and organic matter, it can't be written as soil. Do not write soil.

Line 151-155; the sentence is too long, split it into two sentences.

Line 156;at 16 h light and 8 h dark cycles at 25°C and watered with a sprayer.

Line 156-158; delete the sentence "Plants were watered by wetting the surface....."

Line 160; morphological parameters were measured as described above. "delete which were measured as described in the previous section". And how the dry weight was measured?

Line 162; delete "as described in subsequent sections" and write about how many samples were collected for gene expression and hormone level analysis?

Line 164-166; is the plants were grown again for RNA-seq analysis or just samples were collected from the previous section?

Line 168; write about the parameters to check RNA quality.

Line 173; delete "About 49 M raw reads for each sample were generated" and also for line 175-176, no need to write how many reads were obtained.

Line 163; section Experimental design, RNA extraction, RNA-seq, and analysis, authors must be mentioned about the number of samples collected per replication and how many numbers of samples were subjected for RNA-seq analysis. and also write about the experimental design and treatments.

Line 196; provide the primers information in the supplementary material.

Line 204-207; delete it, it's not necessary

Line 215-216; same as above

Line 216-226; no need to write whole protocol method just, mentioned with citation.

Line 246; vials

Line 264; how much volume of inoculation was used?

Line 286-287; need to revise the sentence as "To initially investigate the growth promotion potential"

Line 294; than control

Line 300; positively influenced the growth of *N. benthamiana*.

Line 349-350; delete the reference in the results, also check in the whole results section. If necessary, move to discussion section

Line 394-397; rearrange the results according to figure

From lines 417-420, there is an unnecessary extension of the sentences. It is better to combine these two sentences.

417-20; MAMPS is not repeated in the text again, so its better to delete it and expended form
In Figure 6, I suggest deleting all other VOCs and only presenting data for 3-octanone and Hexadecane at different concentrations. All the information for other VOCs data can go in supplementary.

VOCs table (Table 1) should be presented as supplementary.

Figure S4 is better to be presented in the main text together with Figure 7.

In lines 433-439, some points related to the previous findings regarding the potential of *Pythium oligandrum* in biocontrol should be moved to the last part of the introduction section.

In lines 472-473, the sentence is not relevant to the study.

Line 538-539; these is long, split into 2-3 sentences

Reviewer #3 (Public repository details (Required)):

RNAseq data needs to be deposited and made available on acceptance of the manuscript

Reviewer #3 (Comments for the Author):

The manuscript is well organized and provides some interesting insight into plant growth promotion and disease mitigation by the biocontrol organism *P. oligandrum*. In general the experiments appear sound and are well explained, it would just be helpful to include all of the data more transparently rather than showing only representative experiments. See specific comments below.

Importance section:

-Highlight the importance of the plant growth promotion for agricultural crops and specifically the crop that this work targets (ginger production)

-Highlight the importance of *P. myriotylum* disease in this crop and why biological control is useful/needed

Materials and methods:

-It would be helpful to have a diagram of the VOC plant growth assay set-up, this would make it easier to visualize how the experiments were conducted and easier to replicate the set-up.

-Specify number of replicates used for the RNAseq

Line 177 - please provide a correct accession number for the RNAseq data

-Was leaf tissue dried before grinding or ground up fresh?

Results:

-For all bar charts show the data points on the graph in addition to just the standard error

-Include the results of all the replicate experiments, this can be either separately in the supplemental material or included all together in the main figures

Staff Comments:

Preparing Revision Guidelines

Please return the manuscript within 60 days; if you cannot complete the modification within this time period, please contact me. If you do not wish to modify the manuscript and prefer to submit it to another journal, please notify me of your decision immediately so that the manuscript may be formally withdrawn from consideration by Microbiology Spectrum.

If your manuscript is accepted for publication, you will be contacted separately about payment when the proofs are issued;

please follow the instructions in that e-mail. Arrangements for payment must be made before your article is published. For a complete list of **Publication Fees**, including supplemental material costs, please visit our website.

Please note that the M&Ms section has been moved to after the Discussion section to meet journal formatting requirements.

Reviewer comments:

Reviewer #2 (Comments for the Author):

In the present study, Sheikh et al showed that the volatile organic compounds produced by a biocontrol agent, *Pythium oligandrum* have the potential to enhance growth and induce resistance against various stress in plants. The authors showed that VOCs produced by *P. oligandrum* also increased the expression of genes for growth-related hormones and various stress responses. From the analysis of the transcriptomic dataset, the authors quantified the growth-related hormones which were in higher concentration in the plants that were exposed to the VOCs, and it supported the growth enhancement in the presence of VOCs. The authors also showed that exposure to the VOCs also reduced the severity of the disease. Overall, the study is comprehensive and well-planned, introduction section is well written, methodology is good, and results are interesting. However, there are some suggestions that should be considered.

Abstract; Line 1; It is better that author write about 1-2 introductory sentences related to the importance of pathogen or disease.

In the abstract, an introductory sentence related to the importance of disease has been expanded according to the suggestion.

Line 35; in sustainable agriculture by enhancing....

The word sustainable has been added to the sentence as suggested by the reviewer.

Line 36-37; is only the effect of VOCs was study related to plant growth not disease suppression? Make it clear write about disease suppression.

Yes, the effect of VOCs was studied on disease suppression as well as plant growth. The words "disease suppression" have been added to the sentence to make it clear for the readers.

Line 37-39; why the tobacco plants were used? As the whole explained about the effect of VOCs on the ginger plant growth and disease severity.

To initially screen the effect of VOCs on plant growth and disease suppression, tobacco plants were used. This has been added in the abstract in line 38.

Lines 45-47; it is better to mention the priming of ginger plants against various stress.

The sentence in line 48 has been modified to mention about the priming of ginger plants against various stress.

Line 45; Write about the conclusion of your findings.

In line 47, the sentence has been modified by adding the words “We conclude that” to the following last sentence in the abstract which we consider a sufficiently concluding statement.

Original sentence:

This study contributes to understanding the growth-promoting mechanisms of *P. oligandrum* on ginger and tobacco the mechanisms of action of *P. oligandrum* as a biocontrol agent.

Revised sentence:

We conclude that this study contributes to understanding the growth-promoting mechanisms of *P. oligandrum* on ginger and tobacco, priming of ginger plants against various stress, and the mechanisms of action of *P. oligandrum* as a biocontrol agent.

Line 51-52; revise the sentence meanings are not clear, write such as "..... produced by *P. oligandrum* related to plant growth promotion and disease resistance".

In Line 55, the words “promotion and disease resistance” has been added to the sentence according to the reviewer’s suggestion.

Original sentence:

This study highlights the significance of VOCs produced by *P. oligandrum* in plant growth as demonstrated by enhanced growth and increased biomass content in the VOCs exposed plants.

Revised sentence:

This study highlights the significance of *P. oligandrum*-produced VOCs in plant growth promotion and disease resistance.

Line 66; delete a non-plant pathogenic "an oomycete"

The reason for writing non-plant pathogenic in Line 66 was to differentiate it from the plant-pathogenic oomycete *P. myriotylum* used in this study. Therefore, it will be better if non-plant pathogenic stays in the sentence to make it easier for the readers to differentiate between the non-plant pathogenic oomycete, *P. oligandrum*, and plant pathogenic oomycete *P. myriotylum*.

Line 69; delete "or by", and induction of host

The word “or by” has been deleted and the text has been changed by adding the word “and” before the following as suggested by the reviewer: “induction of host resistance through elicitors.”

Line 73; delete "recently"

Deleted.

Line 75; as author mentioned *Pythium myriotylum* having a broad host range, so its better to 2-3 examples of diseases caused by *Pythium myriotylum*

Two further examples of diseases caused by *Pythium myriotylum* were added to the text in line 79: damping-off of chilli pepper, and crown and root rot of hemp.

Line 89; author need to check in the whole manuscript, abbreviate the genus name when it appears 2nd time.

This has been checked and corrections have been made according to the reviewer's suggestion.

Line 90-91; revise the sentence

The sentence in line 92 has been modified. We interpret that the confusion could be either caused by the structure or use of the word "also" in the sentence.

original sentence

Individual VOCs have been reported to increase plant growth e.g., 3-octanone, which is also produced by *Streptomyces* spp., increased root growth in *Arabidopsis*

revised sentence

Streptomyces spp., produce 3-octanone which has been reported to enhance root growth in *Arabidopsis*.

In lines 95-99, The sentence structure is not clear, need to revise

The sentence has been rephrased to improve the structure according to the reviewer's suggestion. We understand that the use of the following words "plant pathogenic oomycete" were repeated which made the sentence unclear, as the same words "plant pathogenic oomycete" were also used in the previous paragraph.

original sentence

The plant pathogenic oomycete *P. myriotylum* causes a variety of diseases in a wide range of crops (Plaats-Niterink, 1981), including tobacco (Zhang et al., 2021) and ginger (Daly et al., 2022a)

revised sentence

P. myriotylum has been reported to cause diseases in a wide range of crops (Plaats-Niterink, 1981), including tobacco (Zhang et al., 2021) and ginger (Daly et al., 2022a).

Line 101-104; same as above, need to abbreviate the genus names.

The genus names have been abbreviated.

Original sentence

Interestingly, different plant-pathogenic *Phytophthora* species have been found to produce distinct sets of VOCs. E.g., *Phytophthora plurivora* releases acetoin, 4-hydroxybutanoic acid, α -pinene, and Δ -3-carene, while *Phytophthora cactorum* releases acetone, dimethyl-disulfide, 1-hexanol, 1-heptanol, 1-octen-3-ol, 3-octanone, and 2-octen-1-ol (Loulier et al., 2020).

Revised sentence

Interestingly, different plant-pathogenic *Phytophthora* species have been found to produce distinct sets of VOCs. E.g., *P. plurivora* produces acetoin, 4-hydroxybutanoic acid, α -pinene, and Δ -3-carene, while *P. cactorum* produces acetone, dimethyl-disulfide, 1-hexanol, 1-heptanol, 1-octen-3-ol, 3-octanone, and 2-octen-1-ol (Loulier et al., 2020).

From lines 94-107, in the second last paragraph of the introduction, add some information about the biocontrol oomycete species.

In line 101, according to the reviewer's suggestion, in the second last paragraph of the introduction section, information about two oomycete biocontrol species has been added which is as follows:

As well as *P. oligandrum*, other species of non-plant pathogenic oomycetes have been reported as antagonists of fungal and oomycete plant pathogens and potential biocontrol agents. E.g., *P. nunn* has been reported as a parasite of an oomycete plant pathogen, *P. ultimum* (34), and *P. periplocum* antagonized the fungal plant pathogen *Botrytis cinerea* (35).

In lines 108-114, some points from the first paragraph of the discussion (from lines 433-439) should go into the last paragraph of the introduction.

In lines 115, some points from the first paragraphs of discussion have been repeated in the last paragraph of the Introduction.

Original sentence

In this study, we aimed to investigate if *P. oligandrum* VOCs could contribute to plant growth promotion and disease control.

Revised sentence

In this study, we aimed to investigate if *P. oligandrum* VOCs could contribute to plant growth promotion and disease control apart from previously reported auxin-mediated mechanism of growth promotion and elicitors-induced disease resistance in plants.

Line 120; *P. oligandrum* strain CBS is a plant pathogenic or biocontrol agent?

P. oligandrum CBS 530.74 is a biocontrol strain and is added in the text in line 378.

Line 121; for how long time the strain was grown?

In line 379, general conditions for the growth of all strains were written. Later, for each experiment, it has been explained in each section for the time of growth of strain.

In line 131, it is more useful to be consistent with mentioning the number of plants or seedlings used. Either use it every time in methodology or don't use it.

To be consistent in all sections, the number of plants has also been mentioned in other sections.

Line 132; delete "(grown as described in previous section)"

The following has been deleted "(grown as described in previous section)" according to the reviewer's suggestion.

Line 139-140; number of seedlings per replications?

There were five seedlings in each replicate as mentioned in line 400.

Line 144-145; rewrite the sentence.

The sentence has been re-written for clarity. We interpret that the confusion might be caused by using words "a method based on (Park et al., 2015) was used."

Original sentence

For an *in-planta* evaluation of the effect of *P. oligandrum* GAQ1 VOCs (Po-VOCs) on the growth of *N. benthamiana* and ginger plants, a method based on (Park et al., 2015) was used.

Revised sentence

The effect of *P. oligandrum* GAQ1 VOCs (Po-VOCs) on *N. benthamiana* and ginger in planta was evaluated by following the method described previously (52) and a diagram illustrating the pot-jar assembly used in our study is shown in Figure S10.

In line 153, if it is vermiculite and organic matter, it can't be written as soil. Do not write soil.

All uses of "soil" have been replaced.

Line 151-155; the sentence is too long, split it into two sentences.

The sentence has been split into two according to the reviewer's suggestion.

Original sentence

Equal-sized *N. benthamiana* or ginger seedlings were transferred to the plastic pots containing sterilized vermiculite and organic matter (3:1) and placed on the glass jars containing either *P. oligandrum* or only V8 agar medium as control and firmly sealed by wrapping at least nine times with parafilm (PARAFILM M, Bemis, USA) to avoid the escape of VOCs produced by *P. oligandrum*.

Revised sentences

Equal-sized *N. benthamiana* or ginger seedlings were transferred to the plastic pots containing sterilized vermiculite and organic matter (3:1) and placed on the glass jars containing either *P. oligandrum* or only V8 agar medium as control.

The glass-jar assembly was then firmly sealed by wrapping at least nine times with parafilm (PARAFILM M, Bemis, USA) to avoid the escape of VOCs produced by *P. oligandrum*.

Line 156;at 16 h light and 8 h dark cycles at 25°C and watered with a sprayer.

The sentence has been modified by adding the following words “**and watered with a sprayer**” according to the reviewer’s suggestion.

Line 156-158; delete the sentence "Plants were watered by wetting the surface....."

This sentence was written to explain how the plants were watered and what are the precautions to take for reproducibility as the excess watering caused leakage of water from pots into the jars and resulted in contamination of the media at the bottom of jar. It would be better if this sentence is not deleted.

Line 160; morphological parameters were measured as described above. "delete which were measured as described in the previous section". And how the dry weight was measured?

The sentence has been modified and information for measuring the dry weights has been added according to the reviewer’s suggestion and following sentence has now been added in the text in line 397 as sentence for measuring the dry weight was first time mentioned in the section with this heading “**Effect of *P. oligandrum* VOCs on *N. benthamiana* seedling growth in-vitro**” in the manuscript.

“For measuring the dry weight, plant material was dried in an oven until it reached a constant weight which indicated that all moisture was removed.” .

Line 162; delete "as described in subsequent sections" and write about how many samples were collected for gene expression and hormone level analysis?

The following words "as described in subsequent sections" have been deleted as suggested by the reviewer.

The information about the number of samples have been added in the text in lines 426 as suggested.

For the gene expression and hormone analysis, separate experiments were performed. Three biological replicate leaf samples were collected for each treatment, i.e., three biological replicate samples for VOCs-exposed and three biological replicate samples for non-exposed (control) plants..

Line 164-166; is the plants were grown again for RNA-seq analysis or just samples were collected from the previous section?

The samples for RNA-seq analysis were taken from a separate experiment. In line 425, this has also been added to the text.

Line 168; write about the parameters to check RNA quality.

The parameter to check the quality of RNA samples were to measure the absorbance at wavelengths of A260/280 and A260/230, whereby for all the samples A260/280 was greater than 2.0 and A260/230 was greater than 1.8.

This has been added to the text in lines 437.

Line 173; delete "About 49 M raw reads for each sample were generated" and also for line 175-176, no need to write how many reads were obtained.

The following "About 49 M raw reads for each sample were generated" has been deleted.

In line 446, the information about the number of reads has also been deleted.

Line 163; section Experimental design, RNA extraction, RNA-seq, and analysis, authors must be mentioned about the number of samples collected per replication and how many numbers of samples were subjected for RNA-seq analysis. and also write about the experimental design and treatments.

We apologize for not clarifying the number of replicates for RNA-seq experiment. Following has now been added to the text in lines 433. Part of the experimental design is described and part is referring to the previous section, and we think if the whole of the previous section is repeated here, the M&Ms will be overly repetitive.

“For the RNA sequencing analysis, three biological replicate samples were collected for VOCs-exposed and for non-exposed (control) plants.”

Line 196; provide the primers information in the supplementary material.

The primer sequences used for *N. benthamiana* have been added in the supplemental files. It has also been mentioned and cross-referenced in the text in line 465.

Line 204-207; delete it, it's not necessary

These sentences explain the details of the qPCR program and contain necessary information about the primer specificity. It is therefore better not to delete these lines from the text.

Line 216-226; no need to write whole protocol method just, mentioned with citation.

The method is based on the method in the citation, and for clarity we prefer to write the method we used here.

Line 246; vials

The typo has been corrected.

Line 264; how much volume of inoculation was used?

The inoculum was two autoclaved wheat seeds which had been cultured with *P. myriotylum* for 7 d. We describe this in our previous cited publication.

Line 286-287; need to revise the sentence as "To initially investigate the growth promotion potential

In lines 124, the following word "**growth promotion**" has been added in the text according to the suggestion of the reviewer.

Line 294; than control

The sentence has been modified by adding the words "**than control**" in the text in lines 132 according to the suggestion of the reviewer.

Line 300; positively influenced the growth of *N. benthamiana*.

The sentence has been modified by adding the following "**of *N. benthamiana***" to the text in lines 138 according to the suggestion of the reviewer.

Line 349-350; delete the reference in the results, also check in the whole results section. If necessary, move to discussion section.

Here, it was important to explain what the Pfam domains are and what is the function of each domain. To explain the Pfam domains to highlight the importance of choosing the highly upregulated genes annotated with the described Pfam domains, it was necessary to cite the articles. So, we suggest to keep the citations in this section for better understanding of the readers.

Other than those that are necessary, there are no other similar examples of citations present in the Results section.

Line 394-397; rearrange the results according to figure

In line 229, the results have been re-arranged according to the Figure whereby the parts referring to the ABA data at 21 d were moved after the 7 d and 14 d ABA data.

From lines 417-420, there is an unnecessary extension of the sentences. It is better to combine these two sentences.

The sentences in the lines 252-255 have been modified and combined.

Original sentence

Moreover, the GC-MS chromatograph of *P. myriotylum* SWQ7 showed that both hexadecane and 3-octanone were not detected. Standards were also used to confirm the identification of hexadecane and 3-octanone (Figure S3).

Revised sentences

Moreover, the GC-MS chromatograph of *P. myriotylum* SWQ7 showed that both hexadecane and 3-octanone were not detected and standards were also used to confirm the identity of both VOCs (Figure S8) .

417-20; MAMPS is not repeated in the text again, so its better to delete it and expended form

The word MAMPs has been deleted and written in extended form in the sentence according to the suggestion of reviewer.

In Figure 6, I suggest deleting all other VOCs and only presenting data for 3-octanone and Hexadecane at different concentrations. All the information for other VOCs data can go in supplementary.

The data for all the VOCs in Figure 6 has been moved to supplementary files and the supplementary figure presenting the data of 3-octanone and hexadecane at different concentrations has been moved to main text according to the suggestions of the reviewer.

VOCs table (Table 1) should be presented as supplementary.

The VOCs table has been moved to a supplemental table.

Figure S4 is better to be presented in the main text together with Figure 7.

Figure S4 is now presented in the main text together with Figure 7.

In lines 433-439, some points related to the previous findings regarding the potential of *Pythium oligandrum* in biocontrol should be moved to the last part of the introduction section.

Information has been moved to the last paragraph of the introduction section on the auxin-mediated mechanism of growth promotion and elicitors-induced disease resistance in plants.

In lines 472-473, the sentence is not relevant to the study.

The sentence was deleted (line 303).

Line 538-539; these is long, split into 2-3 sentences.

The sentence has been split into two:

Original sentence

Transcriptomic analysis showed that exposure to Po-VOCs induced the differential expression of putative ginger genes for growth promotion, resistance and hormone signalling which was reflected in the altered hormone concentrations in the leaves of ginger, suggesting the involvement of hormones in signalling pathways important for VOCs-mediated growth promotion and reduction of disease severity in ginger.

Revised sentences

Transcriptomic analysis showed that exposure to Po-VOCs induced the differential expression of putative ginger genes for growth promotion, resistance, and hormone signalling.

The increased hormone concentrations in the leaves of ginger further suggested the involvement of hormones in signalling pathways important for VOCs-mediated growth promotion and reduction of disease severity in ginger.

Reviewer #3 (Public repository details (Required)):

RNAseq data needs to be deposited and made available on acceptance of the manuscript.

The RNA seq data has been deposited to NCBI and GEO accession number is GSE235182.

Reviewer #3 (Comments for the Author):

The manuscript is well organized and provides some interesting insight into plant growth promotion and disease mitigation by the biocontrol organism *P. oligandrum*. In general the experiments appear sound and are well explained, it would just be helpful to include all of the data more transparently rather than showing only representative experiments. See specific comments below.

Importance section:

-Highlight the importance of the plant growth promotion for agricultural crops and specifically the crop that this work targets (ginger production)

In Line 51, part of a sentence has been added in the importance section to highlight this. We did not specifically mention ginger as ginger is emphasised in other places,

“Plant growth promotion plays vital role in enhancing production of agricultural crops and ...”

-Highlight the importance of *P. myriotylum* disease in this crop and why biological control is useful/needed

The importance of *P. myriotylum*-caused disease in ginger and the reason why biocontrol is useful has been explained and the following sentence has been added in the line 61.

“In ginger, rhizome rot disease caused by *P. myriotylum* results in severe losses and biocontrol has a role as part of an integrated pest management strategy for rhizome-rot disease of ginger.”

Materials and methods:

-It would be helpful to have a diagram of the VOC plant growth assay set-up, this would make it easier to visualize how the experiments were conducted and easier to replicate the set-up.

The diagram below showing the set-up of plastic-pot over glass-jar assembly has been added as **Figure S10**.

Figure S10. Pot-jar assembly used in the current study to see the effect of *P. oligandrum*-produced VOCs on plant growth, showing plastic pots with six small holes (2-mm) at the bottom to allow the plants to be exposed to the VOCs produced by *P. oligandrum*. Two layers of filter papers were placed inside at the bottom of the pot to avoid the leakage of contaminating liquid through the holes into the glass jars containing *P. oligandrum* grown on VS agar medium. Plants were grown in the plastic pots containing vermiculite and organic matter (3:1) and fitted onto the glass jar to allow the plants to be exposed to the Po-VOCs. The glass jar assembly was sealed with Parafilm at least 9 times to avoid the escape of VOCs.

-Specify number of replicates used for the RNAseq

We apologize for not mentioning the number of replicates for RNA-seq analysis. Following has been added to the text in lines 433

“For the RNA sequencing analysis, three biological replicate samples were collected for VOCs-exposed and for non-exposed (control) plants.”

Line 177 - please provide a correct accession number for the RNAseq data

The accession number for the RNA seq data is GSE235182.

-Was leaf tissue dried before grinding or ground up fresh?

No, the leaf tissues were not dried before grinding. This has been added to the manuscript on Line 484.

Results:

-For all bar charts show the data points on the graph in addition to just the standard error

Most of the figures in the main text and supplementary figures have been changed to show the individual data points on the graphs in addition to the standard error. The disease index bar charts already showed all the data points.

-Include the results of all the replicate experiments, this can be either separately in the supplemental material or included all together in the main figures.

All the data of the repeat experiments have been shown as supplementary figures, and have also been cross-referenced in the main text as well as in the figure captions.

June 26, 2023

Dr. Paul Daly
Jiangsu Academy of Agricultural Sciences
Institute of Plant Protection
No. 50 Zhongling Street
Nanjing 210014
China

Re: Spectrum01510-23R1 (Volatile organic compounds emitted by the biocontrol agent *Pythium oligandrum* contribute to ginger plant growth and disease resistance)

Dear Dr. Paul Daly:

Your manuscript has been accepted, and I am forwarding it to the ASM Journals Department for publication. You will be notified when your proofs are ready to be viewed.

Sincerely,

Lindsey Burbank
Editor, Microbiology Spectrum
